# The Evaluation of Carbon Farming Strategies in Organic Vegetable Cultivation

**Dan Ioan Avasiloaiei** [1,*] , **Mariana Calara** [1] , **Petre Marian Brezeanu** [1] , **Nazim S. Gruda** [2] **and Creola Brezeanu** [1,*]

1   VRDS Bacău—Vegetable Research and Development Station, Calea Bârladului, 220, 600388 Bacău, Romania; calara.mariana@legumebac.ro (M.C.); brezeanumarian@legumebac.ro (P.M.B.)
2   INRES—Institute of Crop Science and Resource Conservation, Department of Horticultural Science, University of Bonn, Auf dem Hügel 6, 53121 Bonn, Germany; ngruda@uni-bonn.de
*   Correspondence: avasiloaiei_dan_ioan@yahoo.com (D.I.A.); creola.brezeanu@yahoo.com (C.B.)

**Abstract:** The urgent need to mitigate greenhouse gas (GHG) emissions has prompted the exploration of various strategies, including the adaptation of carbon farming practices, to achieve sustainability in agricultural systems. In this research, we assess the viability of carbon farming practices for organic vegetable growing in Europe. The study explores the potential benefits of these practices, including GHG emissions' mitigation and improved soil health, biodiversity, and ecosystem services, while also acknowledging the need for further research to optimize implementation strategies and foster widespread adoption. However, the suitability and effectiveness of carbon farming practices in organic vegetable production systems remain uncertain. The analysis considers the measurement and estimation methods employed to assess changes in soil carbon stocks and the potential environmental and economic implications for farmers. Despite a substantial body of data demonstrating the sustainable attributes of carbon farming and its multifaceted advantages, a degree of hesitancy persists. Considering this, we propose undertaking a concise strengths, weaknesses, opportunities, and threats (SWOT) analysis to evaluate multiple aspects of carbon farming. The findings reveal that carbon farming practices can be viable and advantageous in organic vegetable production. Carbon farming practices, such as cover cropping, reduced tillage, compost application, and agroforestry, can significantly enhance the sustainability of organic farming systems. Implementing these practices can mitigate greenhouse gas emissions, improve soil health and fertility, and promote biodiversity conservation. Farmer education and support, policy measures, and continued research are crucial for maximizing the potential of these practices for a sustainable future. These practices also contribute to developing climate-friendly agricultural systems, promoting environmental resilience, and reducing the ecological footprint of organic vegetable production. However, further research is needed to optimize implementation strategies, address site-specific challenges, and foster widespread adoption of carbon farming practices in organic vegetable production.

**Keywords:** soil carbon storage; soil fertility; ecological vegetable system; GHG mitigation; regenerative practices

## 1. Introduction

Currently, humanity can only effectively address environmental and climate change issues if it fully embraces ecological principles for genuine sustainability. However, history has repeatedly shown that we tend to take substantial action only during crises and that we often fall short of complete commitment. In this framework, the widely endorsed DNSH (Do No Significant Harm) principles, which have recently gained popularity among political factors, signify nothing more than a return to fundamental ecological values by striving to "not significantly harm" the environment.

Carbon, as an essential element, plays a fundamental role in life on Earth. It forms the building blocks of human DNA and is pervasive in our food [1]. However, using carbon-based fossil fuels over the past century for energy generation and industrial processes has led to the accumulation of greenhouse gas (GHG) emissions, primarily carbon dioxide ($CO_2$), in the atmosphere. This accumulation has caused various detrimental effects, including global climate warming, biodiversity loss, increased ocean acidity, and severe meteorological phenomena (droughts, heatwaves, floods, wildfires, severe thunderstorms, mudslides, and landslides) [2].

Therefore, the situation has become critical, and it has led to the imposition of extreme measures. At the European level, this means the achievement of zero greenhouse gas emissions by 2050, with an intermediate stage aiming at a 55% decrease by 2030 compared to the currently recorded value, according to the EU Climate Law [3]. It is said that farming represents the "single biggest cause" of the worst air pollution not only in Europe but also in China, the US, and Russia. Hence, restrictive measures regarding a series of agricultural practices need to be applied, and a new approach in farming techniques and technologies has thus become imperative. Paustian et al. and Mattila et al. [4,5] stressed the crucial role of carbon sequestration in agricultural soils to mitigate climate change.

Carbon farming encompasses a range of agricultural techniques (cover cropping, no-till farming, agroforestry, crop rotation, organic farming, use of perennial crops, managed grazing, wetland restoration, use of biochar, compost and mulching, reforestation and afforestation) designed to capture atmospheric carbon and store it within the soil and in crop roots, wood, and leaves. Additionally, Tariq et al. [6] define carbon farming as a comprehensive approach to maximize the uptake of carbon dioxide from the atmosphere and enhance its storage in plant matter and soil organic matter across working landscapes. This holistic approach involves implementing practices that are proven to accelerate the removal of $CO_2$ from the atmosphere and facilitate its sequestration in vegetation and soil.

A study conducted by Fageria [7] illustrates that the intake of soil organic matter and composition, which is inextricably linked with the amount of carbon stored at the soil level and water retention capacity, is impacted by management practices. A direct correlation between soil organic matter and its ability to retain water while enhancing structure and productivity has been outlined by several studies [8–10], with an inverse relationship between drought and disease occurrence [9,11,12]. Certain studies [13,14] have emphasized the significance of promoting agricultural practices that facilitate organic matter sequestration in the soil, thus reducing environmental $CO_2$ levels. Carbon farming is pivotal in aligning with new ecological standards concerning climate change resilience and impact [4,5,15,16].

Considering these factors, our study aimed to investigate and establish a symbiotic connection between ecological vegetable cultivation methods and carbon farming practices. This objective stems from the inherent similarities in the operational principles of both approaches, which emerged naturally and highlighted the potential for synergies between these two critical fields.

## 2. Materials and Methods

*Systematic Literature Review*

In this review article, we performed a comprehensive systematic literature review (SLR) following the guidelines outlined in the Preferred Reporting Items for Systematic Reviews and Meta-Analysis (PRISMA) methodology (Table 1). Our main objective was to systematically review and synthesize the existing literature on carbon farming strategies in organic vegetable cultivation, assess their impact on carbon sequestration, soil health, and crop productivity, and identify gaps in the research. Mainly, the present study was developed by means of no fewer than nine questions of paramount importance for the subject, as follows: (1) How have carbon farming strategies been defined and conceptualized in the context of organic vegetable cultivation? (2) What are the primary objectives and goals of implementing carbon farming strategies in organic vegetable cultivation? (3)

What are the key carbon sequestration practices employed in organic vegetable cultivation? (4) How do carbon farming strategies impact soil health parameters? (5) How do carbon farming practices influence crop productivity (yield) and quality (nutrient content, pest resistance) in organic vegetable cultivation? (6) What economic benefits or costs are associated with the adoption of carbon farming strategies in organic vegetable cultivation? (7) What are the key barriers and challenges faced by farmers when implementing these strategies? (8) What areas require further investigation to enhance our understanding of these strategies? (9) How do government policies and regulations impact the adoption and effectiveness of carbon farming strategies in organic vegetable cultivation?

**Table 1.** Data source and selection activities according to PRISMA methodology.

| Phase Number | Activity Description |
|---|---|
| Phase 1 | Research database identification using Web of Science, Scopus, Google Scholars, ScienceDirect, MDPI, and Springer.com |
| Phase 2 | Assessment of the research papers with relevance for the subject published in prominent journals |
| Phase 3 | Removal of papers that were found irrelevant to the subject or could cause scientific incoherence |
| Phase 4 | Draft the actual paper, including the relevant literature |

Some of the keywords used to find the relevant scientific papers were: carbon farming, carbon sequestration, carbon in vegetable organic farming, and soil management in organic farming. In a subsequent stage, a qualitative data analysis was performed using Covidence systematic review software (Veritas Health Innovation, Melbourne, Australia. Available online www.covidence.orgmanagement accessed on 20 August 2023)–studies were identified, analyzed, and ranked, focusing on their content and thematics to ensure their significance and relevance to the selected topic. With this in mind, we conducted a systematic literature review, examining 273 scientific papers related to the subject (Table 2).

**Table 2.** Keywords used and the number of scientific papers generated.

| 1st Keyword Used | 2nd Keyword Used | Number of Research Papers | |
|---|---|---|---|
| | | Initial Search | After Filtering |
| Carbon farming | | 855 | 46 |
| Carbon sequestration | | 1054 | 73 |
| Organic biochar | | 723 | 21 |
| Organic conservation tillage | | 162 | 7 |
| Agroforestry | | 136 | 6 |
| Organic cover crops | | 398 | 19 |
| Organic nutrient management | | 103 | 11 |
| Organic soil management | Vegetable cultivation | 908 | 34 |
| Permaculture | | 50 | 11 |
| Organic carbon footprint | | 206 | 6 |
| Urban farming system | | 218 | 12 |
| Conservation farming | | 309 | 14 |
| Regenerative practices | | 188 | 11 |
| Zero budget farming | | 7 | 2 |
| Total number of research papers | | 5317 | 273 |

The studies selected for this synthesis met the following criteria: to be relevant so that the data provided has practical applicability—given the novelty degree of the subject studied, this condition is implied; all the data presented have a solid scientific background; and the opinions of the authors are well argued and find approval within the scientific community.

Furthermore, to highlight the potential feasibility of carbon farming measures practical to organic vegetable cultivation, we conducted a SWOT analysis, highlighting the four essential aspects: strengths, weaknesses, opportunities, and potential threats.

Similarly, to highlight the trend of specific aspects such as emissions generated by human activities, particularly in agriculture, or the areas occupied by organic farming at the European level, relevant databases for these fields were consulted, such as Statista, European Environment Agency, or IFOAM Organics Europe.

## 3. Results and Discussion

### 3.1. Overview

Our goal is to evaluate the potential of carbon farming to reduce agricultural greenhouse gas emissions, including carbon dioxide, methane, and nitrous oxide, which are significant contributors to climate change. By implementing sustainable land management practices, maximizing carbon sequestration, and minimizing carbon emissions, we consider carbon farming as an opportunity to mainly transform the organic vegetable cultivation system into effective carbon sinks and contribute to a more sustainable future.

Efforts to combat GHG emissions should prioritize a combination of mitigation and adaptation strategies [17,18]. This involves shifting towards low-carbon energy sources, enhancing energy efficiency, advocating for sustainable transportation and production methods, improving waste management approaches, and implementing nature-based solutions into action in order to manage pollution in key industrial sectors, as shown in Figure 1. International cooperation and collaboration are vital to accelerate the deployment of clean technologies and knowledge sharing to achieve significant emission reductions [19].

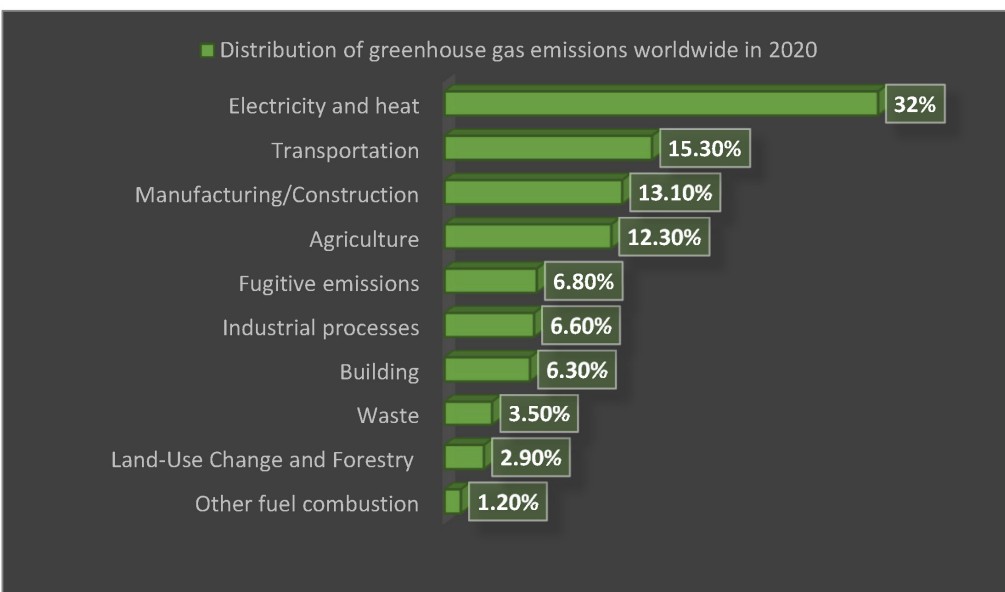

**Figure 1.** Distribution of greenhouse gas emissions worldwide in 2020 by sector (https://www.statista.com/statistics/241756/proportion-of-energy-in-global-greenhouse-gas-emissions/, accessed on 23 June 2023).

Equity and fairness are crucial factors to be considered in the scientific examination of global greenhouse gas (GHG) emissions [20]. Developed countries bear a historical

responsibility due to their substantial contributions to cumulative emissions, necessitating supporting developing nations as they transition towards low-carbon economies [21].

The sources of agricultural emissions are diverse and significant, contributing to the global challenge of climate change (Figure 2). Livestock production, particularly enteric fermentation and manure left on pasture, is a significant source of agricultural greenhouse gas emissions. Addressing this issue requires promoting efficient feed conversion, adopting improved manure management systems, and exploring alternative protein sources [22].

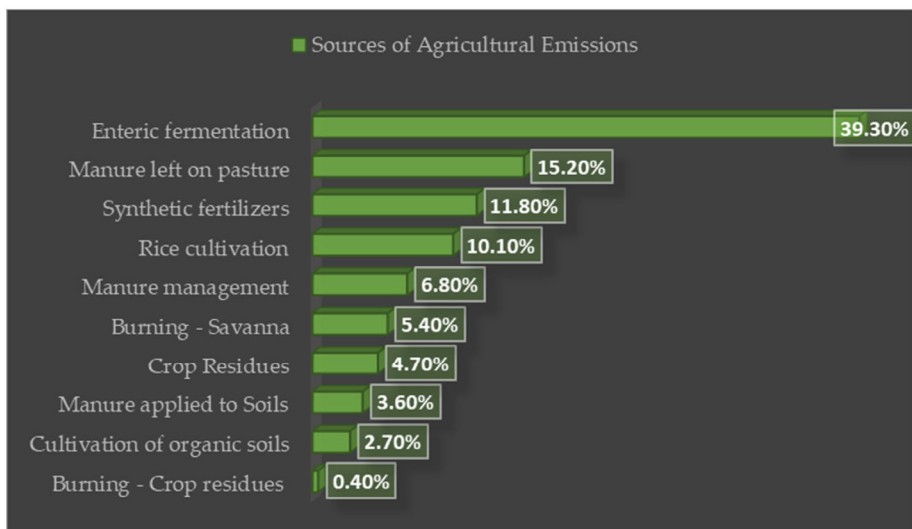

**Figure 2.** Sources of Agricultural Emissions (https://www.wri.org/insights/5-questions-about-agricultural-emissions-answered, accessed on 12 July 2023).

Furthermore, at the European level, the emissions from agriculture represent a significant concern in terms of environmental impact and climate change (Figure 3). Policy frameworks at the regional and national levels and financial incentives could play a crucial role in driving the adoption of sustainable agricultural practices [23]. Riccaboni et al. [24] emphasize the preeminence of supporting research and innovation in agriculture, facilitating knowledge exchange among farmers, and raising awareness among consumers about the environmental impact of their food choices.

Thus, Verschuuren [25] highlights the significant progress that can be made in reducing greenhouse gas emissions and contributing to building a more resilient and sustainable agricultural sector by addressing emissions from agriculture at the European level. Fytili and Zabaniotou [26] discuss the need for a holistic and collaborative approach involving farmers, policymakers, researchers, and consumers to successfully transition towards a low-carbon and environmentally responsible agriculture industry.

GHG net emissions/removals by land use, land use change, and forestry (LULUCF) encompass alterations in atmospheric concentrations of all greenhouse gases linked to changes in forests and land use practices, comprising, yet not restricted to, (1) the discharge and sequestration of $CO_2$ resulting from variations in biomass stocks due to forest administration, logging, fuelwood collection, etc.; (2) the transformation of existing forests and natural grasslands into alternative land uses; (3) the sequestration of $CO_2$ resulting from the abandonment of previously managed lands (e.g., croplands and pastures); and (4) $CO_2$ emissions and removals [27].

Organic agriculture has experienced significant expansion at the European level, driven by consumer demand for healthier and more sustainably produced food (Table 3). The sector's core principles, including prohibiting synthetic pesticides and fertilizers, promote biodiversity conservation, soil health, and reduced environmental impact. These practices can potentially mitigate greenhouse gas emissions, protect water resources, and preserve natural habitats [28].

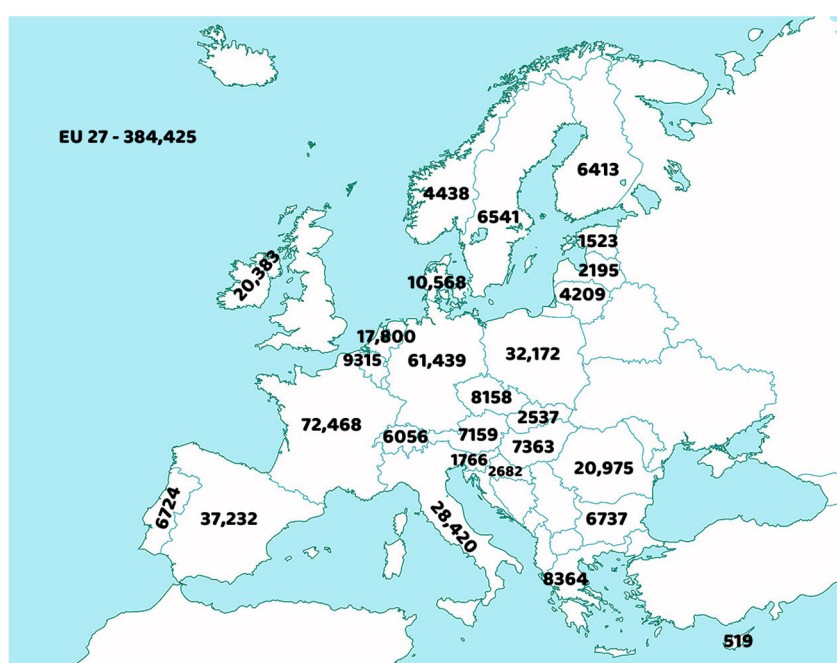

**Figure 3.** GHG projections with existing measures (WEM scenario) from the agricultural sector (kilotons of $CO_2$ equivalents) at the EU-27 level in 2023 (excluding LULUCF) (https://www.eea.europa.eu/data-and-maps/data/data-viewers/eea-greenhouse-gas-projections-data-viewer, accessed on 2 July 2023).

**Table 3.** Organic agriculture in 2021 at the European level (adaptation based on IFOAM Organics Europe).

| COUNTRY | Percentage of Organic Agricultural Land (%) | Organic Land Area (1000 Hectares) |
|---|---|---|
| Liechtenstein | 40.20 | 1 |
| Austria | 26.50 | 679 |
| Estonia | 23.00 | 227 |
| Sweden | 20.20 | 607 |
| Switzerland | 17.40 | 181 |
| Italy | 16.70 | 2186 |
| Czech Republic | 15.80 | 558 |
| Latvia | 14.80 | 291 |
| Finland | 14.40 | 328 |
| Slovakia | 11.70 | 223 |
| Denmark | 11.40 | 300 |
| Germany | 10.80 | 1802 |
| Spain | 10.80 | 2635 |
| Slovenia | 10.80 | 52 |
| Greece | 10.10 | 535 |
| EU-27 | 9.60 | 15,600 |
| France | 9.60 | 2777 |
| Lithuania | 8.90 | 262 |
| Croatia | 8.10 | 122 |
| Portugal | 7.80 | 308 |
| Belgium | 7.40 | 102 |
| Hungary | 5.90 | 294 |
| Cyprus | 5.70 | 8 |
| Luxembourg | 5.20 | 7 |
| Norway | 4.60 | 45 |
| Romania | 4.30 | 579 |
| Netherlands | 4.20 | 76 |

**Table 3.** *Cont.*

| COUNTRY | Percentage of Organic Agricultural Land (%) | Organic Land Area (1000 Hectares) |
|---|---|---|
| Poland | 3.50 | 509 |
| United Kingdom | 2.80 | 489 |
| Ireland | 1.90 | 87 |
| Montenegro | 1.70 | 4 |
| Bulgaria | 1.70 | 86 |
| Turkey | 0.90 | 328 |
| Serbia | 0.70 | 24 |
| Republic of North Macedonia | 0.60 | 8 |
| Malta | 0.60 | 0.1 |
| Kosovo | 0.50 | 2 |
| Iceland | 0.40 | 6 |
| Bosnia and Herzegovina | 0.10 | 2 |
| Albania | 0.10 | 1 |

The vegetable cultivation sector has a two-fold impact on the environment. Firstly, it contributes to climate change (CC) through various activities (soil tillage, fertilization, irrigation, fossil fuel consumption). Secondly, the resulting environmental changes, in turn, affect vegetable production. Thus, a reciprocal relationship exists between horticultural activities and their ecological consequences, impacting the climate and fresh food production sector [17,18].

Organic farming (OF) embraces a comprehensive strategy for agricultural operations and food cultivation, emphasizing eco-friendly and climate-conscious methodologies, preserving natural resources, the fostering of biodiversity, and adhering to stringent standards of animal welfare and production [29,30]. Regarding the eco-friendly practices, OF prioritizes environmentally friendly methods by avoiding the use of synthetic pesticides, herbicides, and chemical fertilizers. Instead, it relies on natural alternatives and sustainable practices to manage pests, weeds, and soil fertility. In terms of the climate-conscious methods, OF aims to mitigate climate change by reducing greenhouse gas emissions and promotes practices such as crop rotation, cover cropping, and reduced tillage, which help sequester carbon in the soil and enhance soil health. For the preservation of natural resources, OF places a strong emphasis on protecting natural resources such as soil and water, promoting practices that prevent soil erosion, improve soil structure, and conserve water resources. Furthermore, OF encourages biodiversity in agricultural landscapes. This includes planting diverse crop varieties, creating habitats for beneficial insects and wildlife, and avoiding monoculture farming, which can be detrimental to biodiversity. In organic livestock farming, animals are typically raised under higher welfare standards compared to conventional methods. This includes providing access to the outdoors, pasture grazing, and adhering to strict regulations regarding animal health and well-being. Not least, OF adheres to strict production standards and certification processes. These standards encompass everything from soil management and pest control to animal husbandry and food processing. Third-party organizations certify farms and products as organic to ensure compliance [29,30].

Lal [31] defines carbon sequestration and carbon farming extensively. Carbon sequestration involves transferring carbon dioxide ($CO_2$) from the atmosphere and long-lasting, secure storage at the soil level by improving the accumulation of organic and inorganic carbon stocks. This is achieved through the adoption of appropriate land use practices and a set of recommended management techniques, such as mulch farming [32,33], conservation tillage [34], agroforestry [35], diverse cropping systems [36–39], cover crops [40,41], and integrated nutrient management, including the use of manure, biosolids, compost, sustainable forest management, and improved grazing [42]. The sequestration of soil organic carbon (SOC) is influenced by agricultural systems that promote the incorporation of significant amounts of biomass into the soil, minimize soil disturbance, protect soil and

water resources, enhance soil structure, increase the activity and diversity of soil organisms, and strengthen elemental cycling processes.

Thus, carbon farming is presently a prominent aspect of sustainable agricultural practices due to its climate-related advantages, primarily through carbon sequestration in agricultural soils [43,44]. Soil carbon levels are closely linked to soil organic matter [45], which significantly influences the soil's structure, health, and nutrient content. Alterations in soil organic matter resulting from climate change and modifications in management approaches can impact water retention capabilities.

A series of very well-documented studies highlighted the preeminence of the organic farming system in terms of carbon sequestration at the soil level. Thus, Gattinger et al. [46] revealed that over 14 years, the soil organic carbon stocks in the upper 20 cm were $3.50 \pm 1.08$ Mg C ha$^{-1}$ higher in organic than in nonorganic systems. In the same way, Leifeld and Fuhrer [47] determined in their review an average annual increase in the SOC concentration in organic systems of 2.2% compared to conventional methods, where it did not vary significantly. Furthermore, using calculations based on the combination of single practices, such as extensification, improved rotations, residue incorporation, and manure use, Freibauer et al. [48] assessed the C sequestration potential of organic farming in Europe to be 0–500 kg ha$^{-1}$ y$^{-1}$.

Recently, Palayukan et al. [49] emphasized that the implementation of ecological farming methods resulted in notable enhancements in the physical and chemical characteristics of the soil. Additionally, these practices increased the accessibility of the organic carbon fraction. Likewise, Sardiana [50] demonstrated the preeminence of an ecological farming system in terms of carbon sequestration, highlighting an annual increase of 1.13 tons per hectare compared to conventional farming practices. Thus, most findings emphasize the organic farming potential for increasing C stocks in agricultural soils [31,51,52].

Moreover, Tuomisto et al. [53] reported a distinct correlation between organic vegetable growing and external carbon inputs, which were higher than the conventional system. Blair et al. [54] and Tejada et al. [55] certified an improved soil structure among the apparent benefits.

However, adopting a holistic approach when considering organic materials in agriculture is essential. Although organic materials can influence soil's physical and chemical properties, it becomes crucial to address the specific type and origin of these materials. For instance, peat is commonly used as an organic horticultural substrate, but its production involves the destruction of peatlands, which has profound environmental and climate change implications [56,57]. Therefore, it is imperative to carefully evaluate the sustainability and ecological consequences of using organic materials in horticultural practices.

Furthermore, considering a comprehensive approach that addresses the interplay between soil, plants, and the environment is of utmost importance. Recently biostimulants and biofertilizers, such as plant-growth-promoting rhizobacteria (PGPR) and arbuscular mycorrhizal fungi (AMF), have gained popularity in horticulture due to their positive effects on plant nutrition [58–60]. Utilizing the natural activity of soil microbes, biostimulants hold immense potential for breaking down toxins in the soil and harnessing the in situ soil microbial activity [61]. Similarly, biofertilizers can promote plant growth by aiding nitrogen fixation and phosphorus solubilization. Additionally, they synthesize plant hormones such as indole acetic acid and cytokinins and promote gibberellin synthesis within plants [62,63]. Moreover, they contribute organic acids and enzymes, enhancing water and nutrient uptake and bolstering systemic resistance against diseases [64,65]. Optimizing the synergistic interactions between soil and environmental factors, along with implementing good agricultural practices, including the strategic utilization of effective plant-beneficial microorganisms in the root zone, epitomizes the concept of smart agriculture. This integrated framework effectively harnesses the potential of these combined approaches [66–68].

### 3.2. Environmental Impact of Organic Vegetable Cultivation on Carbon Emissions

Regarding the carbon footprint (CF) in organic vegetable production, Adewale et al. [69] conducted a study on a small organic vegetable farm. They highlighted soil emissions, irrigation, fuel use, and organic fertilizer as the main hotspots. A viable option in this matter is replacing gasoline and diesel with biodiesel and the alternative of solar-powered irrigation systems instead of grid-powered irrigation systems that could determine a reduction in the farm's CF of 34%. Regarding vegetable crops, cauliflower, potato, and pepper displayed the highest CF $ha^{-1}$.

The organic farming system emits significantly less $CO_2$ per unit of land than the conventional method. On the other hand, when reported on per unit of production, both the organic and the conventional systems demonstrate similar emissions, primarily due to the lower yield levels associated with organic farming [70].

According to Küstermann and Hülsbergen [71], the energy consumption per hectare in the organic farming system is significantly lower, approximately 2.75 times less, compared to the conventional method. Furthermore, the organic system exhibited substantially reduced nitrous oxide (852 $CO_2$ eq $kg^{-1}$ compared with 1307 $CO_2$ eq $kg^{-1}$ recorded in conventional farming) and carbon dioxide emissions (349 $CO_2$ eq $kg^{-1}$ as opposed to 707 $CO_2$ eq $kg^{-1}$), along with significantly higher carbon sequestration rates (+110 kg C $ha^{-1}$ $a^{-1}$ = reduction in the greenhouse potential by 415 kg $CO_2$ eq $ha^{-1}$ $a^{-1}$) [71].

Over the past two decades, numerous studies have examined the environmental impact of agricultural practices on the production of organic vegetables and their carbon footprint. Thus, Smith et al. [72] revealed that organic vegetable production had an energy input of 50% compared to conventional carrots, 65% for onions, and 27% for broccoli. Fritsche and Eberle [73] found that GHG emissions ($CO_2$ eq $kg^{-1}$) were between 15% and 31% lower for organic tomatoes than conventional ones. De Backer et al. [74] stated that an organic leek crop releases only 33% of conventional system emissions. The same trend was recorded for organic potato crops [75–77]. Antithetically, Ziesemer [78] highlighted that organic carrot and potato production had 43% higher energy inputs per unit of output due to mechanical weeding. Overall, Wood et al. [79] showed that organic vegetable production has about 50% of the energy intensity of conventional systems. It was observed that various organic crop rotations generated lower $N_2O$ emissions per hectare compared to conventional rotations. Regarding energy usage for fertilization, a critical aspect of organic vegetable cultivation, the scientific community agrees that applying compost and manure entails relatively low fuel and energy costs [80,81].

Organic farming has a higher potential for carbon sequestration than conventional farming systems [82]. Additionally, organic fertilization, the incorporation of crop residues into the soil, and the cultivation of cover crops have the potential to increase $N_2O$ emissions. However, cultivating deep-rooted crops can help mitigate $NO_3$ leaching [83].

### 3.3. Exploring the Connection between Organic Vegetable Farming and Carbon Sequestration

The exclusion of agrochemicals' utilization with a substantial carbon footprint in organic agriculture (OA) production can yield notable advantages for the climate and natural resources. Synthetic nitrogen (N), phosphorus (P), and potassium (K) fertilizers, pesticides, and other agrochemicals contribute to the deterioration in soil organic matter, leading to its depletion, as well as the pollution of water sources [29], and having a direct contribution to climate change [17,18].

Numerous strategies for increasing carbon sequestration in organic vegetable cultivation have been evidenced by the scientific literature. For example, adopting a perennial vegetable system can have a beneficial impact on mitigating climate change, promoting biodiversity, and improving nutritional value. Regarding carbon sequestration, it is estimated that the large-scale cultivation of perennial vegetables would induce an intake between 22.7 and 280.5 MMT $CO_2$ eq/year by 2050 [84].

Soil management strategies in organic farming have a significant role in global environmental conservation, particularly in increasing soil carbon levels, e.g., by recycling

organic matter. According to Sánchez-Marañón et al. [85], assessing soil organic carbon in organic farming systems is crucial for understanding their impact on carbon sequestration. Other studies have highlighted that organically managed soils tend to have higher soil carbon than conventional farming systems [46,86]. Organic farms have the potential to mitigate greenhouse gas (GHG) emissions due to practices such as legume cultivation for nitrogen supply and reduced reliance on external inputs such as fertilizers and agrochemicals. Hence, organic vegetable cultivation could potentially reduce global GHG emissions by approximately 20% by avoiding industrial nitrogen fertilizers [87]. Furthermore, the potential carbon sequestration capacity of organic farming could offset 40–72% of the world's annual agricultural GHG emissions. These findings have been confirmed by other studies, indicating that organic fertilizers can increase soil carbon content more than chemical fertilizers [88–90].

The utilization of cover crops represents another crucial technological link for enhancing carbon sequestration at the soil level. For instance, Adewale et al. [69] found that about 498 kg C ha$^{-1}$ yr$^{-1}$ were lost from the soil's top 30 cm without cover crops as opposed to 57 kg with cover crops, which means 1826 and 209 kg $CO_2$ eq ha$^{-1}$ yr$^{-1}$, respectively. Additionally, incorporating cover crops significantly reduced estimated soil greenhouse gas (GHG) emissions by 41%.

Using legumes in crop rotation and multiple cropping systems offers the following advantages: it preserves and potentially enhances soil quality, boosts the yields of vegetable crops, and exerts a positive influence on the functioning of the ecosystem [91,92]. Furthermore, Jensen et al. [93] indicated other benefits of cutting N fertilizers, such as mitigating the farmer's costs and environmental risks related to greenhouse gas emission release.

Biochar research is experiencing rapid growth, primarily due to its potential for carbon sequestration [94]. Additionally, biochar holds promise as a technology for immobilizing pollutants [95], waste management, enhancing soil fertility [96], and use as a sustainable growing media and peat replacement [56,97]. Previous studies have linked the impact of biochar on crop yield to various factors, including increased cation exchange capacity and nutrient retention [56,98], raised pH and base saturation [97,99], elevated availability of phosphorus [98], and improved plant-accessible water content [100].

Multiple strategies are available for carbon sequestration in organic vegetable cultivation soils, referred to as "adaptive restorative practices", as outlined in Table 4. These practices include no-tillage or reduced tillage intensity [101], the increased input of crop residues [102], the application of organic manure [103], the utilization of cover crops [104,105], the implementation of organic mulching [32,33], nutrient management [106,107], a reduction in or the elimination of fallow periods [108,109], and the restoration of permanent forests and grasslands [31,110–114]. These strategies are primarily characterized by minimizing soil disturbance and increasing the application of organic matter. Furthermore, Nishimura et al. [115] underlined the effectiveness of the legume intercropping system and crop rotations [116] for soil fertility enhancement, with nitrogen and phosphorous having a pivotal role in the process.

Regarding soil tillage practices in organic vegetable cultivation, some studies suggest the potential of maintaining soil organic carbon by implementing no-tillage techniques [101,117].

An alternative and feasible approach is the adoption of natural farming (NF) [118], a low-input, no-tillage system incorporating weed residue mulching. This system, initially developed by Fukuoka [119], involves cutting weeds using a brush cutter and utilizing the resulting residues as a natural cover for the area.

**Table 4.** Carbon farming measures suitable for ecological vegetable growing.

| Measures | Expected Results | References |
|---|---|---|
| Rethink tillage management | Shifting conventional tillage to no-till farming cuts emissions by 30 to 35 kg C/ha per season | [31] |
| | Higher SOC after 5 years of organic vegetable production by adopting no tillage | [101] |
| | Formation of macroaggregates under long-term conservation tillage | [120] |
| Use of cover crops | Provides nutrients to the soil | [40] |
| | Decreases soil $NO_3$ by 30% | [41,104,121] |
| | C sequestration outweighs $N_2O$ emissions | [105] |
| Crop rotation and no tillage | Higher SOC compared to conventional tillage system | [117,122] |
| | Higher mineralized soil N compared with tilled systems | [123,124] |
| | An overall decline of up to 7.6% in GWP (net global warming potential) | [125] |
| Use of grain crop residues as fertilizers | 1000 kg of cereal residue generates 12 to 20 kg N, 1 to 4 kg P, 7 to 30 kg K, 4 to 8 kg Ca, and 2 to 4 kg Mg | [102] |
| Adapting a natural farming system | low-input NT with weed residue mulching increases soil carbon sequestration by 0–7.5 cm for 8 years | [126] |
| Use of biochar— the carbonaceous product obtained from the organic material pyrolysis | Adjusts soil N cycle and reduces N losses | [127] |
| | An average 63% increase in symbiotic biological dinitrogen ($N_2$) fixation in crops and an 11% enhancement of plant N uptake | [127] |
| | Supplies nutrients to soils that stimulate biological $N_2$ fixation | [128,129] |
| | Enhances crop yields | [130] |
| | Reduces $N_2O$ emissions | [131,132] |
| Soil erosion control | Measurement of emissions compared with burial of C under erosional processes | [72,133] |
| Management of farming practices: - Long-term organic matter application | Mitigate organic matter degradation that impacts the atmosphere similar to fossil fuel combustion | [42] |
| | Enhances soil organic carbon level and fertility | [106,107] |
| | Reduces $N_2O$ emissions | [105,134] |
| | Magnifies free-living N fixation rates in the soil and/or nodulation in N-fixing crops | [135] |
| - Use of silicate rock amendments | Speeds up the rate of chemical weathering and consumes atmospheric $CO_2$ | [136] |
| | Decreases both methane and $N_2O$ emissions | [137,138] |
| Water resources conservation | Water-efficient farming systems = water conservation = water harvesting | [31,139] |
| Carbon trading market scope | Short-term (3–5 years) measure of proficiency of SOC level changes | [140,141] |

*3.4. Advantages and Disadvantages of Carbon Farming: Exploring the Upsides and Downsides*

Many research studies [142–146] have highlighted several co-benefits of carbon farming, including but not limited to the promotion of food and nutritional security, the purification and renewal of water resources, the enhancement of biodiversity, and the restoration of degraded soils and ecosystems.

Several meta-analyses comparing organic and conventional farming systems have consistently concluded that organic farming systems exhibit superiority in soil organic matter content [46,53,147].

Organic vegetable systems are foremost in terms of carbon sequestration due to the utilization of organic amendments and cover crops [148,149], as well as the implementation of more diverse crop rotations [150,151]. It is essential to underline that these technological aspects are not limited solely to certified organic systems.

From a financial standpoint, carbon markets offer farmers the opportunity to generate additional income by implementing recommended management practices (RMP) that focus on sequestering soil organic carbon (SOC) and reducing emissions [152,153].

All of the recommended practices for carbon farming offer clear environmental advantages. For instance, adopting no-till practices promotes the formation of soil aggregates, facilitating long-term storage of carbon while reducing $CO_2$ emissions associated with disturbance [154]. Cover cropping enhances both above- and below-ground plant biomass, thereby increasing carbon inputs to the soil [154]. Although it can contribute to carbon storage in both above- and below-ground biomass, agroforestry may compete with crops for land unless intercropped [155]. Various soil amendments, such as biochar, silicate rock, and organic amendments, are gaining momentum as carbon sequestration practices. While they may have similar practical uses, their pathways for carbon sequestration differ significantly. Biochar, for example, adds and stabilizes carbon in the soil, silicate rock amendments accelerate weathering processes, and organic amendments enhance crop productivity and subsequent carbon formation in the soil [127,136].

Organic vegetable cultivation practices are distinguished by reduced reliance on synthetic inputs, resulting in lower usage of chemically synthesized products and a decreased demand for primary energy compared to conventional systems [66].

The vegetable farmers were willing to adapt practices such as retaining crop residue, using no-till techniques, and applying organic mulch [142]. Gruda [33] reported that organic mulch can improve water retention, balance soil temperature, increase vegetable growth, and reduce weed growth by withdrawing light [32]. Field and greenhouse trials confirmed these results, showing up to 63% fewer weeds and significant growth improvements in crops such as head lettuce and sugar melons, even when it was impossible to suppress all kinds of weeds [32]. In terms of potential drawbacks, it is crucial to highlight the findings from the studies of Leifeld and Fuhrer [47] and Powlson et al. [156] that suggest the beneficial impact of organic vegetable cultivation on soil organic carbon content may be attributed to the higher application of organic fertilizers compared to conventional systems. Consequently, increasing soil organic carbon content may not necessarily represent actual carbon sequestration.

An additional concern is represented by lower average yields recorded in organic vegetable cultivation systems. This implies that a larger land area would be required to ensure a comparable availability of energy and protein for human consumption in the scenario of a global shift to 100% organic agriculture by 2050 [157]. A similar viewpoint was discussed by Smith et al. [158], who highlighted the necessity for increased land use to compensate for the predicted 40% lower yields in organic production. However, this is only sometimes realizable. Converting more natural habitats for agriculture could lead to deforestation and the loss of biodiversity, thereby compromising delicate ecosystems [159]. Moreover, such an approach may exacerbate land degradation and soil erosion issues [160]. In regions with limited arable land, expanding cultivation areas might prove impractical and could trigger conflicts over land ownership [161]. In addition, expanding organic farming through increased land use might have socioeconomic implications [162]. Small-scale farmers, who predominantly practice organic agriculture, could face challenges acquiring more land due to escalating costs and limited availability [160]. Large-scale land conversion might lead to the displacement of local communities, altering their traditional ways of life and threatening their livelihoods [163].

A distinct correlation exists between the sequestration of organic soil carbon and the corresponding nitrogen intake. According to Hungate et al. [164], approximately 10 g of organic carbon requires around 1 g of additional nitrogen. Thus, the potential consequences of increased $N_2O$ emissions and $NO_3^-$ leaching from agricultural soils in organic systems due to the heightened demand for additional nitrogen fertilizers should be considered and further evaluated [108].

Another constraint is that, on a production unit basis, the energy use and carbon footprint do not frequently favor organic systems [165,166]. A summary of the main potential advantages and disadvantages carried out as a SWOT analysis is presented in Table 5, as evidenced by numerous reviewed studies.

**Table 5.** SWOT analysis of carbon farming practices for ecological vegetable cultivation.

| STRENGTHS | References | WEAKNESS | References |
|---|---|---|---|
| The organic vegetable system uses less energy and stores higher values of C per hectare | [53,166,167] | On a production unit bases, both energy use and carbon footprint are higher in organic vegetable growing | [165,166] |
| Reduced soil erosion improves soil structure and water quality, and reduces sedimentation | [168,169] | Carbon lasting/persistence and stabilization | [170] |
| Reducing tillage minimizes the use of irrigation water by increasing soil water-holding capacity | [168,169] | Political, economic, and social factors | [171] |
| Improves water quality and ecology | [168,169,172,173] | C sequestration may lead to a cut in food and fiber production causing higher food prices and reducing exports. | [168,169] |
| Soil health upgrade | [174,175] | Heightened $N_2O$ emission | [105,176] |
| Food safety | [177] | | |
| Public health welfare | [178] | | |
| Carbon sequestration at the soil level is a process characterized by three features: naturalness, environment friendly, and cost-effective | [31,141,179] | | |
| Enhances soil $CH_4$ oxidization capacity | [180–182] | | |
| **OPPORTUNITIES** | **References** | **THREATS** | **References** |
| Integration of SOC monitoring and Carbon footprint into the organic farming certification process | [183] | Policymakers and farmers' divergent targets and interests | [184] |
| Use of stacking environmental credits so that payments overcome costs | [185] | Unattractive carbon contracts | [186,187] |
| Facilitation of carbon farming contracts for leased farmland | [188–190] | Land ownership and the challenge of farm leasing | [185] |
| Farmers could receive payments from the government grounded in the area of land enrolled in a GHG mitigation program | [191–193] | | |
| Biodiversity preservation | [194] | | |
| Poverty mitigation | [195] | | |
| Infrastructure upgrading | [196] | | |

As a future perspective, Adewale et al. [183] suggest that including carbon footprint and soil organic carbon monitoring in the organic farming certification process would increase compliance with the final rule and upgrade the available information about carbon sequestration and the carbon footprint of organic agricultural systems and practices.

*3.5. The Synergy between Organic Vegetable Cultivation and Some Emerging Systems within the Context of Carbon Farming*

Recently, there has been an increasing interest in developing innovative agricultural systems prioritizing sustainability, productivity, and environmental stewardship. Several low-input agricultural systems, including permaculture, urban gardening, agroforestry, conservation agriculture, and Zero Budget Natural Farming (ZBNF), exhibit genuine potential for transforming our approach to food production, emphasizing efficiency, ecological balance, and resilience. These innovative approaches offer promising solutions for reducing greenhouse gas emissions, sequestering carbon, and enhancing overall environmental sustainability.

Urban gardening can significantly reduce the carbon footprint of food production and distribution by enabling local cultivation. This approach minimizes transportation emissions and lowers energy consumption, which is commonly associated with long-distance food supply chains [197].

Urban gardening, combined with organic farming, represents a sustainable and environmentally friendly approach that encompasses various benefits. This practice promotes local food production, improves food security, and contributes to climate change mitigation. Moreover, it addresses public health issues and provides a means of adapting to geopolitical challenges in urban areas. Overall, urban gardening and organic farming are integrated solutions supporting multiple sustainability aspects in urban environments [29].

The agroforestry system increases organic matter accumulation in soil surface residues. It promotes more effective conservation of biodiversity. Sequestering carbon in trees and soil contributes to mitigating $CO_2$ emissions and addresses the challenges posed by climate change [35].

Recent studies on perennialization highlight that the intentional integration of perennial species can have positive effects on various ecosystem services, including provisioning (agricultural yields), regulating (pest control, hydrological cycles, water quality, carbon sequestration, and storage), and supporting (soil quality, pollination) services [198]. Furthermore, permaculture's focus on enhancing yield through beneficial interactions has anticipated the emergence of the functional diversity field, where ecologists now refer to this phenomenon as overyielding driven by complementarity or facilitation [199].

A brief description of the main characteristics of low-input systems is presented in Table 6.

**Table 6.** New types of low-input vegetable growing systems and the carbon farming approach.

| A New Type of Vegetable Cultivation System | Synergy with Organic Vegetable Cultivation | Expected Outcomes | References |
|---|---|---|---|
| Urban farming | Make use of recycled materials sourced from the local area, such as compost produced from bio-waste | Alleviates environmental footprint and actively contributes to the development of a sustainable bioeconomy | [39] |
| | Supply green areas within urban environments | Adjusts temperatures, mitigates air pollution, and enhances air quality, thereby fostering healthier and sustainable urban environments in response to climate change | [37] |
| Conservation farming | Precision farming and minimum tillage, crop rotation, and residue retention | Enhances crop yield in sandy, acidic soils | [200] |
| | Preservation and restoration of crucial soil characteristics, including organic carbon content, structure, and biological diversity and activity | Initiates a soil quality restoration process that addresses microbiological activity and soil fauna diversity | [34] |

| A New Type of Vegetable Cultivation System | Synergy with Organic Vegetable Cultivation | Expected Outcomes | References |
|---|---|---|---|
| Agroforestry system | Integration of trees into annual food crop systems (both perennial and annual species): trees and vegetable crops | Tree leaves offer sufficient nutrients for building and sustaining soil fertility and supplying nutrients to plants | [35,201] |
| | | Enhance carbon storage in both above-ground and below-ground components | |
| Zero Budget Natural Farming (ZBNF) | Natural resources and implementation of diverse cropping systems, and utilizing products derived from cow dung and urine to enhance soil biology | Significant positive contribution to preserving the ecosystem and mitigating the detrimental effects caused by agrochemicals | [38] |
| Permaculture | The pursuit of enhancing beneficial connections between elements to attain optimal design and harness their synergistic potential | Minimizes waste, human labor, energy, and resource inputs while establishing systems that maximize benefits and achieve high holistic integrity and resilience | [36,202] |

The emergence of new low-input agricultural systems reflects society's growing recognition of the urgent need for sustainable and resilient food production. Permaculture, urban gardening, agroforestry, conservation agriculture, and Zero Budget Natural Farming are just a few innovative approaches that promise to transform the agricultural landscape. Integrating ecological principles, minimizing external inputs, and prioritizing long-term sustainability contribute to a more food-secure, environmentally friendly, and socially inclusive future. Embracing and further developing these approaches will pave the way for a more resilient, ecologically conscious, and carbon neutral agricultural sector.

The presented study should be perceived as consistent with conventional vegetable production. The era of pitting these approaches against each other has passed. Smith et al. [158] conducted a life-cycle assessment to evaluate the impact of a complete shift to organic food production in England and Wales on net greenhouse gas (GHG) emissions. Findings suggest significant deficiencies in the production of most agricultural products compared to a conventional baseline. While organic farming reduces direct GHG emissions, compensating for domestic supply shortfalls through increased overseas land use results in higher net emissions. To effectively address climate change, combining both approaches with a prioritization of sustainable practices would be beneficial. These two approaches have the potential to complement each other greatly. While this has been true in the past, we believe it will continue to be the case.

To summarize, Figure 4 presents an antithetical exposition of the main recommended management practices (RMPs) for carbon farming in contrast to the principal polluting elements from agriculture.

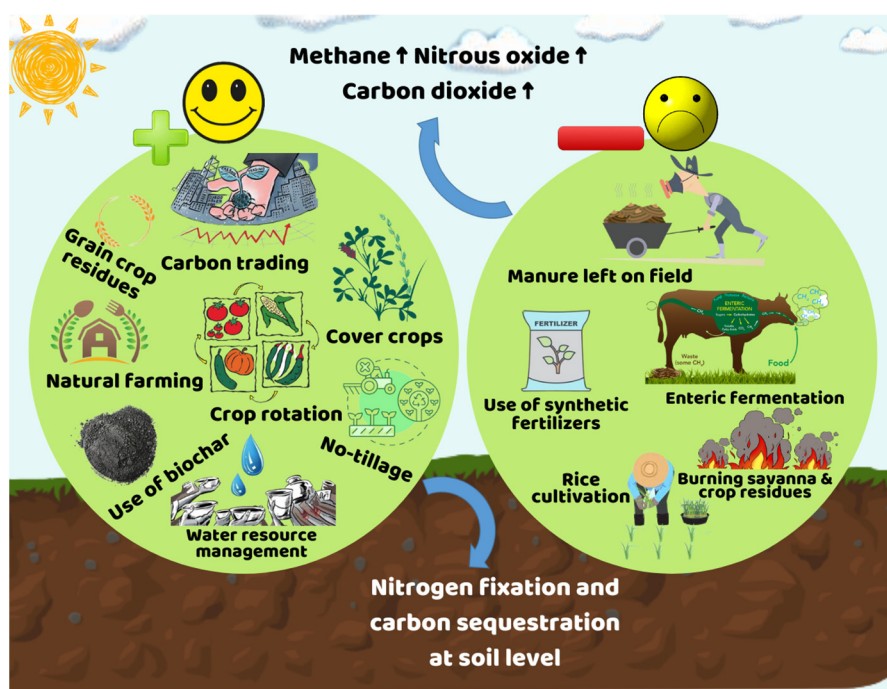

**Figure 4.** The influence of carbon farming recommended management practices compared to the main agricultural polluters.

## 4. Conclusions

Carbon farming practices demonstrate significant potential for enhancing the sustainability of organic farming systems. Implementing these practices can contribute to carbon sequestration, mitigate greenhouse gas emissions, and improve overall soil health and fertility. Incorporating cover cropping, reduced tillage, compost application, and agroforestry in organic farming can effectively increase carbon sequestration rates. These practices enhance soil organic matter content, promote nutrient cycling, and support organic agricultural systems' long-term productivity and resilience.

Carbon farming practices in organic farming systems mitigate climate change impacts and offer co-benefits such as improved water infiltration, reduced soil erosion, and increased biodiversity. These practices enhance ecosystem services through biodiversity conservation and contribute to the sustainability of organic farming, promoting habitat diversity, supporting beneficial organisms, and strengthening the overall ecological balance of agricultural landscapes.

Further improving soil structure and moisture retention, these practices help mitigate the effects of extreme weather events, such as droughts and floods, on crop productivity.

The sustainability benefits of carbon farming practices extend beyond the farm level. By sequestering carbon in soils, organic farming systems contribute to reducing atmospheric carbon dioxide levels and addressing global climate change challenges.

Adopting carbon farming practices requires careful consideration of site-specific factors such as soil type, climate, and available resources. Tailoring these practices to local conditions and organic farming systems is crucial for maximizing their effectiveness and sustainability. The successful implementation of carbon farming practices in organic farming relies on farmer education and support, as well as access to technical expertise and financial incentives. Policy measures and collaborative efforts among stakeholders are essential for promoting the widespread adoption of these practices.

Long-term monitoring and research are necessary to assess the persistence and stability of carbon sequestration achieved through carbon farming practices in organic farming. Continued scientific investigation will provide insights into these practices' scalability, economic viability, and environmental benefits. Integrating carbon farming practices into

organic farming represents a promising pathway towards sustainable agriculture. These practices align with organic farming principles, enhance soil carbon stocks, and contribute to climate change mitigation and adaptation goals.

In conclusion, scientific evidence underscores the potential of carbon farming practices to enhance the sustainability of organic farming systems. These practices offer a valuable approach to achieving a more sustainable and resilient agricultural future by sequestering carbon, improving soil health, and providing additional ecological benefits. Efforts should focus on knowledge dissemination, policy support, and further research to optimize the implementation and maximize the potential of carbon farming practices in organic farming for a sustainable future.

**Author Contributions:** Conceptualization, D.I.A., P.M.B. and M.C.; methodology, D.I.A. and M.C.; software, D.I.A.; validation, P.M.B., M.C. and C.B.; formal analysis, D.I.A.; investigation, D.I.A., M.C. and P.M.B.; resources, D.I.A., M.C., P.M.B. and C.B.; data curation, D.I.A.; writing—original draft preparation, D.I.A.; writing—review and editing, D.I.A., N.S.G. and C.B.; visualization, M.C.; supervision, P.M.B.; project administration, P.M.B. and D.I.A.; funding acquisition, P.M.B. All authors have read and agreed to the published version of the manuscript.

**Funding:** This research received no external funding; the authors acknowledge the in-kind and cash support of the Vegetable Research and Development Station (VRDS), Bacau, Romania.

**Data Availability Statement:** No new data were created in this study. Data sharing is not applicable to this article.

**Acknowledgments:** The authors acknowledge the Sectorial Plan of the Romanian Ministry of Agriculture and Rural Development, implemented by VRDS Bacau through ADER projects during 2023–2026 for its objectives aiming to improve and implement climate-friendly agricultural systems, to promote environmental resilience, and to reduce the ecological footprint of vegetable production discussed in this review.

**Conflicts of Interest:** The authors declare no conflict of interest.

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
