# Peer review of "The Evaluation of Carbon Farming Strategies in Organic Vegetable Cultivation"

_agronomy, doi:10.3390/agronomy13092406_

Round 1

Reviewer 1 Report

Thank you very much for making this manuscript (The Evaluation of Carbon Farming Strategies in Organic Vegetable Cultivation) available for my consideration. Review papers are interesting and time-consuming. The research is interesting and has a very interesting scientific appeal. I emphasize that the text elements must be standardized, and in the results the incorporation of numerical values is crucial.

My point-by-point observations are highlighted below:

Line 4: Change “and” to “comma”.

Line 12: Use “carbon”

Line 20 and 23: Use only “carbon farming”.

- General comment of the Abstract:

Please clarify the purpose of the research and conclusion.

Line 51: In the sentence “severe meteorological phenomena”, please add examples of some of these phenomena.

Line 56: I believe "single biggest cause" would be better

Line 59-60: What were they? Please highlight this in the text, it is an important finding!

Line 78: Does your study have any hypotheses? What motivated this type of research?

Line 85: What was the methodology that was used in the work? Did you rely on any specific methodology?

Line 91: In the "identified, analyzed, and ranked" step, was any website, software, manager used? How was the work with “duplicate” removed?

Was the job search limited to Europe?

Your methodology deserves one more level of detail, please add more details to it.

Line 132: What does LUCF mean?

Line 134: How significant is this? Please add numerical values in the findings.

Line 277: Is it CO2e kg−1 or CO2 eq kg−1?

Line 281: How much bigger?

Line 306: The number 2 is subscripted.

Line 495: I think that before its conclusion, a type of graphic abstract (https://cdn.wikifarmer.com/wp-content/uploads/2023/03/carbon-farming.png) that encompasses the results of all searches would fit. This would make the text more illustrative and eye-catching.

- General comment from the Conclusions section:

While emphasizing good ideas, please put some numerical values.

Some revisions will be needed.

Author Response

Thank you for your suggestions. We have incorporated your recommendations into the draft. All of them are also described one by one below.

Line 4: Change “and” to “comma”. - AMENDED

Line 12: Use “carbon” - AMENDED

Line 20 and 23: Use only “carbon farming”. – AMENDED

- General comment of the Abstract:

Please clarify the purpose of the research and conclusion - please see the manuscript

Line 51: In the sentence “severe meteorological phenomena”, please add examples of some of these phenomena. - AMENDED

Line 56: I believe "single biggest cause" would be better- AMENDED

Line 59-60: What were they? Please highlight this in the text, it is an important finding! - AMENDED

Line 78: Does your study have any hypotheses? What motivated this type of research? -IN THE PENULTIMATE PARAGRAPH BEFORE THE CONCLUSIONS, THERE IS AN EXPOSITION OF OUR HYPOTHESIS AS WELL AS A SERIES OF SUBJECTIVE CONSIDERATIONS. WE CONSIDERED THIS APPROACH OPPORTUNE IN ORDER TO FIRST PRESENT THE RESULTS OF THE SCIENTIFIC COMMUNITY AND ONLY THEN, BASED ON THEM, EXPRESS OUR OWN OPINIONS

Line 85: What was the methodology that was used in the work? Did you rely on any specific methodology? –NO. THE BASIC METHODOLOGY THAT WE HAVE DESCRIBED IT

Line 91: In the "identified, analyzed, and ranked" step, was any website, software, manager used? How was the work with “duplicate” removed? - AMENDED

Was the job search limited to Europe? NOT SPECIFICALLY

Your methodology deserves one more level of detail, please add more details to it.

Line 132: What does LUCF mean? - AMENDED

Line 134: How significant is this? Please add numerical values in the findings.-  WE HAVE CONSIDERED PERCENTAGES TO BE RATHER CONCLUSIVE

Line 277: Is it CO2e kg−1 or CO2 eq kg−1? - AMENDED

Line 281: How much bigger? - AMENDED

Line 306: The number 2 is subscripted. - AMENDED

Line 495: I think that before its conclusion, a type of graphic abstract (https://cdn.wikifarmer.com/wp-content/uploads/2023/03/carbon-farming.png) that encompasses the results of all searches would fit. This would make the text more illustrative and eye-catching. - AMENDED.  A graphical abstract was developed and inserted (see Figure 4 on pag 17).

Thank you once again. We believe that your suggestions have enormously improved the quality of the paper.

Reviewer 2 Report

The Evaluation of Carbon Farming Strategies in Organic Vegetable Cultivation

L-11: It is better to write “including the adaptation of carbon farming practices” instead of “including of adapting carbon farming practices”

L-27: Replace “contribute to” with “contribute in”.

L-36-44: “Nowadays, all the downsides humanity faces regarding the influence of its activities on the environment and their impact on climate change would have been manageable, the critical condition being the assimilation of ecological principles, which determines sustainability stricto sensu. Unfortunately, as humanity has proven countless times throughout history, it only acts in extreme situations, and often, even then, it does not fully commit itself. What else do the highly promoted DNSH (Do No Significant Harm) principles represent, very popular among the political factor in the last few years, if not a return to the fundamental values of ecology by "not significantly harming" the environment?”- Message and sentence structure is not clear. Rewrite the sentences in a simple way so that the readers can understand the meaning.

L-121: promoting sustainable transportation and production- not clear, rewrite it

Figure 3: Need to clarify the values inside the figure.

L-182-183: “Vegetable cultivation sector has a two-fold impact on the environment. Firstly, it contributes to climate change (CC) through various activities.”- Needs to clarify the activities that contributes to climate change.

L187-190: Discuss how organic farming embraces a comprehensive strategy for agricultural operations and food cultivation, emphasizing eco-friendly and climate-conscious methodologies, preserving natural resources, the fostering of biodiversity, and the adherence to stringent standards of animal welfare and production.

L263: CF ha−1. What is CF? Before using the abbreviated form, firstly mention both full form and abbreviated form together.

L264-267: “The organic farming system emits significantly less CO2 per (unit??) area than the conventional method [70]. On a per unit of production basis, both the organic and the conventional systems demonstrate similar emissions, primarily attributed to the lower yield levels associated with organic farming.”- Is there any reference for the second sentence? – Findings of the sentences are contradictory. Check carefully.

L-270-272: “Furthermore, the organic system exhibited substantially reduced nitrous oxide and carbon dioxide emissions, along with significantly higher carbon sequestration rates.”- Is there any values regarding reduced nitrous oxide and carbon dioxide emissions & increased carbon sequestration rates due to organic farming? It is suggested to include numerical values for nitrous oxide emission, carbon dioxide emission & carbon sequestration rates under the organic system and traditional farming system.

L-425: Instead of NO3-, it is better to write NO3-

Other comments: For soil health improvement, organic farming plays very vital role. Therefore, it is suggested to make a table highlighting the effects of different organic farming practices on soil health properties.

Quality of English is up to the marked level. However, minor editing will improve the quality of the manuscript.

Author Response

Respected Reviewer,

Thank you very much for taking the time to review this manuscript. We have incorporated your recommendations into the draft, and we resubmitted.

Please find below the detailed responses for each suggestion of you:

L-11: It is better to write “including the adaptation of carbon farming practices” instead of “including of adapting carbon farming practices” -  AMENDED;

L-27: Replace “contribute to” with “contribute in”. – AMENDED;

L-36-44: “Nowadays, all the downsides humanity faces regarding the influence of its activities on the environment and their impact on climate change would have been manageable, the critical condition being the assimilation of ecological principles, which determines sustainability stricto sensu. Unfortunately, as humanity has proven countless times throughout history, it only acts in extreme situations, and often, even then, it does not fully commit itself. What else do the highly promoted DNSH (Do No Significant Harm) principles represent, very popular among the political factor in the last few years, if not a return to the fundamental values of ecology by "not significantly harming" the environment?”- Message and sentence structure is not clear. Rewrite the sentences in a simple way so that the readers can understand the meaning – AMENDED;

L-121: promoting sustainable transportation and production- not clear, rewrite it – AMENDED

Figure 3: Need to clarify the values inside the figure - ALTHOUGH THE NUMBERS MAY APPEAR CROWDED WHEN INSERTED INTO THE TEXT, IN THE FINAL SIZE WITHIN THE MDPI TEMPLATE, WE CONSIDER THE FIGURES ARE MORE THAN LEGIBLE

L-182-183: “Vegetable cultivation sector has a two-fold impact on the environment. Firstly, it contributes to climate change (CC) through various activities.”- Needs to clarify the activities that contributes to climate change. – AMENDED;

L187-190: Discuss how organic farming embraces a comprehensive strategy for agricultural operations and food cultivation, emphasizing eco-friendly and climate-conscious methodologies, preserving natural resources, the fostering of biodiversity, and the adherence to stringent standards of animal welfare and production. – AMENDED

L263: CF ha−1 . What is CF? Before using the abbreviated form, firstly mention both full form and abbreviated form together.  – AMENDED

 L264-267: “The organic farming system emits significantly less CO2 per (unit??) area than the conventional method [70]. On a per unit of production basis, both the organic and the conventional systems demonstrate similar emissions, primarily attributed to the lower yield levels associated with organic farming.”- Is there any reference for the second sentence? – Findings of the sentences are contradictory. Check carefully. – AMENDED

L-270-272: “Furthermore, the organic system exhibited substantially reduced nitrous oxide and carbon dioxide emissions, along with significantly higher carbon sequestration rates.”- Is there any values regarding reduced nitrous oxide and carbon dioxide emissions & increased carbon sequestration rates due to organic farming? It is suggested to include numerical values for nitrous oxide emission, carbon dioxide emission & carbon sequestration rates under the organic system and traditional farming system. – AMENDED

L-425: Instead of NO3-, it is better to write NO3 –– AMENDED

Other comments: For soil health improvement, organic farming plays very vital role. Therefore, it is suggested to make a table highlighting the effects of different organic farming practices on soil health properties – WE FULLY AGREE WITH THE ASSERTION THAT SOIL HEALTH IMPROVEMENT IS OF PARAMOUNT IMPORTANCE, ESPECIALLY IN THE CASE OF ORGANIC AGRICULTURE, WITH A DIRECT CORRELATION BETWEEN THEM. HOWEVER, TABLE 4 PRIMARILY ADDRESSES THE IMPACT OF GOOD PRACTICES IN ORGANIC FARMING ON SOIL HEALTH SO WE CONSIDERED THAT INTRODUCING A NEW TABLE FOR EXCLUSIVELY ADDRESSING THIS TOPIC MIGHT RISK DUPLICATING CERTAIN INFORMATION.

Thank you once again.

We believe that your suggestions have enormously improved the quality of the paper.

Reviewer 3 Report

Greenhouse gas emissions mitigation is a largely approached, far from solving matter in worldwide research nowadays as it is an important contributor to climate change which heavily impacts human livelihood. The paper holistically looks at it from the organic vegetable farming point of view – another point of interest in ensuring food security.

The Introduction clearly and reasoning presents the background for the research and its aims.

The research methods were thoroughly organized, documentation was elaborate, and they are strictly presented in the respective chapter.

The manuscript is well structured which gives it more clarity an makes it easier to read.

The discussions bring into balance various aspects of the different agricultural systems and highlight both benefits and shortcomings which gives a real objective view on the carbon farming practices.

This comprehensive overview can be a point of reference for future research in the domain.

Author Response

Respected Reviewer,

Thank you very much for taking the time to review this manuscript.

Thank you for your conclusions considering this comprehensive overview as a point of reference for future research in the domain.

Round 2

Reviewer 1 Report

Thank you very much for making a corrected version of the work available. It appears that the authors did not highlight the changes made to the text. This is very important.

Unfortunately, no changes were made to the abstract as requested and the authors responded that this had been done. This is a lack of respect for the reviewer.

Although the suggestions have not been marked, the work has quality to follow. The other suggestions were met. However, the Methodology section still needs more details. Please detail the steps in more detail. This can be found in some works in the literature on this topic. A detailed methodology will assist in future work by other experts in the field.

Little adjustments.

Author Response

Dear Reviewer,

Thank you for your thoughtful and constructive feedback on our revised manuscript. We appreciate your time and effort in evaluating our work. We have taken your comments seriously and would like to address each of your points individually:

Highlighting Changes: We apologize for not highlighting the changes made to the text in the revised manuscript. We understand the importance of transparency in the revision process. We considered, based our previous experience our answer for each specified row provides clarity to our feedback. We understand now it was not enough, and clear markers (in blue) for all modifications in Abstract and Material and Methods are included in the final version.

Abstract Revisions: We acknowledge that no changes were made to the abstract as per your request in the initial review. We did not AMENDED this part, we just asked for a reevaluation.  The requested revision to the abstract is carried out in the final version. Please accept it was not an oversight on our part, and we sincerely apologize for any inconvenience this may have caused.

Quality of Work: We are pleased to hear that you found the overall quality of our work to be acceptable despite some of the issues raised. Your feedback has been invaluable in improving our manuscript.

Methodology Section: We appreciate your recommendation to provide more detailed information in the Methodology section. The changes are highlighted in blue.

Once again, we want to express our gratitude for your thorough review and dedication to upholding academic research standards. We addressed your concerns and provided a revised manuscript that meets the highest quality standards.

Sincerely,

Creola Brezeanu on behalf of all authors
